# Temporal trend of population structure, burden of diseases, healthcare resources and expenditure in China, 2000–2019

Zhenguo Liang [ID],[1] Dongze Wu [ID],[1,2] Cui Guo,[3,4] Jieruo Gu[1]

ZL, DW and CG contributed equally.

[1]Department of Rheumatology, Third Affiliated Hospital of Sun Yat-Sen University, Guangzhou, Guangdong, China
[2]Department of Rheumatology and Immunology, Sichuan Provincial People's Hospital, Chengdu, Sichuan, China
[3]Jockey Club School of Public Health and Primary Care, The Chinese University of Hong Kong, Hong Kong SAR, China
[4]Department of Urban Planning and Design, Faculty of Architecture, The University of Hong Kong, Hong Kong SAR, China

**Correspondence to**
Dr Jieruo Gu;
gujieruo@mail.sysu.edu.cn

## ABSTRACT

**Objectives** To explore the evolutionary trend of population structure, disease burden, healthcare resources and expenditure in China, and to identify key domains that are most in need of intervention.

**Design** A cross-sectional and longitudinal analysis.

**Data source** Population and healthcare data from China Statistical Yearbook, and disease burden attributable to causes and risk factors from the Global Burden of Diseases between 2000 and 2019.

**Measures and methods** We used the Joinpoint Regression Program to measure trends in population composition, population change, dependency ratio, healthcare institution, personnel, expenditure and disease burden from 2000 to 2019.

**Results** Regarding the population in China between 2000 and 2019, a decreasing trend was observed among youth aged 0–14 years (average annual percent change (AAPC): −1.17, a slow rising trend was observed among individuals aged 15–64 years (AAPC: 1.10) and a rapidly increasing trend was observed among individuals older than 65 years (AAPC: 3.67). Astonishing increasing trends in healthcare institutions (AAPC: 3.97), medical personnel (AAPC: 3.26) and healthcare expenditures (AAPC: 15.28) were also observed. Among individuals younger than 70 years, neoplasms (AAPC: 0.54) and cardiovascular diseases (AAPC: 0.67) remained among the top three causes, while tobacco (AAPC: 0.22) remained a top three risk factor. However, while musculoskeletal disorders (AAPC: 1.88) were not a top three cause in 2000, they are a top three cause in 2019.

**Conclusion** Comprehensive age/cause/risk factor-specific strategies are key to reconcile the tension among the triad of population ageing, disease burden and healthcare expenditure. The disease burden from cardiometabolic diseases, neoplasms and musculoskeletal disorders was identified as key domains that require intervention to reduce an increasing disease burden among individuals currently older than 70 years, as well as those approaching this age group.

## STRENGTHS AND LIMITATIONS OF THIS STUDY

⇒ This study is the first to comprehensively characterise the patterns and temporal trends in population structure and composition, health resources and expenditures, life expectancy and disease burden in China from 2000 to 2019.

⇒ This study does not only provide an overview of population, healthcare and disease burden in China, but also inform the policymakers to pay more attention to the increasing age-driven disease burden in older population.

⇒ Although data from the Global Burden of Disease and China Statistical Yearbook are well refined, the quality of the data source needs to be further improved.

⇒ The principal limitations of this study are the unavailability of provincial-level data in disease burden, population and healthcare, and inability to provide granular policy recommendation.

## INTRODUCTION

Together with the world's population, people in China struggle against cancer and ageing despite efforts to extend lifespan and maximise healthspan.[1 2] Since 1950, the lifespan at birth has been extended to 26 years in China, while the total fertility rate has decreased from 5.91 to 1.43.[3] Between 2000 and 2019, the life expectancy for males has increased from 69.1 to 74.4 years, while the life expectancy for females has increased from 73.9 to 80.8 years.[3] It has been forecasted that China will become the largest economy by 2035, and that the population in China will decline by 48.0% from 2017 to 2100.[4] Moreover, in the next 20 years, the universal two-to-three-child policy may not reverse a shrinking workforce and rapid population ageing, especially in rural areas.[5 6] In 2009, China expanded its social health insurance coverage to its entire population and reformed its healthcare delivery system.[7 8]

Healthcare systems coordinate complex inter-relationships among individuals who receive and provide medical financial support.[9] It would have been difficult for China to rigorously control the two pandemics that occurred in 2003 and 2019 without a healthcare system. In particular, an increasing number of modernised medical institutions and well-educated personnel have been established over the past two decades

in China.[10 11] Accelerating rural-to-urban migration and corresponding increases in urban populations have also both led to improved access to healthcare, although substantial health risks can accompany changes in human activities, diets and social structures.[12] Although two Global Burden of Diseases (GBD) studies have assessed the disease burden and risk factor at provincial level in China,[13 14] this study further evaluates healthcare professionals and expenditure and identifies key domains to mitigate the age-driven disease burden.

The ageing society in China continues to exacerbate the burden borne by current family and public healthcare systems, which make it necessary to make long-term strategic plans to respond to the pressures of an ageing society at the governmental, individual and technological level.[15] There are very low spatial and temporal matching degrees between the ageing population rate and the number of medical resources per thousand residents in China, and the geographical pattern of the temporal matching between them exhibited a feature of north–south differentiation.[16] China does not have a typical disease characteristic of an ageing society comparing the characteristics of the ageing population in China with those in the world.[17] Namely, China faces the dual threat of non-communicable diseases (NCDs) and communicable diseases, and the former account for most of the age-related disease burden. Although there are a great number of studies centring on trend of population, burden of disease and healthcare in China, they failed to dissect the implication of a central tension among three mutually conflicting forces: intractable disease burden, population ageing and upsurging healthcare expenditure. This study tries to find a solution to reconcile the three contradictory forces, achieving longer healthy lifespan in the ageing society at high-cost performance of health expenditure. Possible driving forces of disease burden among ageing populations were also examined to provide a basis for targeted interventions, such as early disease prevention and health management among older populations.

## METHODS
### Data sources
The National Bureau of Statistics (NBS) of China annually publishes the China Statistical Yearbook (CSY).[18] The Department of Population and Employment Statistics of the NBS conducts statistical population surveys as follows. A national population census is conducted in years ending with 0; a national 1% population sample survey is conducted in years ending with 5 and sample surveys on population changes are conducted in the remaining years. The latter covers about one per thousand of the total population of China. The sample surveys on population change take the entire nation of China as the population, while each province, autonomous region or municipality represents a subpopulation. A stratified multistage systematic probability-proportional-to-size

cluster sampling scheme is used. Data regarding population, healthcare institutions, personnel, beds and health expenditures are described in detail in the online supplemental appendix (including a glossary).

The GBD 2019 study assessed life expectancy, disease burden, risk factor, aetiology and impairment.[3 19 20] All-cause incidence, prevalence, death, years lived with disability, years of life lost, disability-adjusted life years (DALYs) and attributable risk factors were assessed between 1990 and 2019. The study identified 22 level 2 leading causes and 20 level 2 leading risk factors. Both causes and risk factors were assessed according to number, rate and age-standardised rate of DALYs. The estimation methods and data for burden of disease published in GBD 2019 were used to assess disease burden in China between 2000 and 2019 for the present study.

Aggregated data of population and healthcare from CSY and aggregated data of life expectancy, disease burden attributable to 22 causes or 20 risk factors from GBD between 2000 and 2019 were used to quantify temporal trend of population structure, burden of diseases, healthcare resources and expenditure in China.

### Outcomes of measurements
The primary outcomes are temporal trend in population (population composition, population change, dependency ratio), healthcare (number of healthcare institutions, number of health personnel, healthcare expenditure), life expectancy (healthy and total), disease burden attributable to 22 causes or 20 risk factors (DALYs) from 2000 to 2019. The secondary outcomes are (1) the main drivers of disease burden from 22 causes or 20 risk factors in individuals younger and older than 70 years; and (2) temporal trend in population, healthcare and disease burden attributable to 22 causes or 20 risk factors from 2000 to 2009, and from 2010 to 2019. Driver refers to the cause or risk factor that has the greatest impact on the total disease burden attributable to 22 causes or 20 risk factors from 2000 to 2019.

### Age groups of population and disease burden
Two sets of age group were used to specifically assess the trend of population composition and disease burden. Population composition (0–14, 15–64, 65+ years) and dependency ratio (gross, children, old) were used to reflect the trend of working population and population ageing as age 0–14, 15–64 and 65+ group were younger population, working population, post-working population, respectively. The analysis of disease burden in population younger and older than 70 years aimed to identify the cause or risk factor (driver) which has the greatest impact on the trend of disease burden attributable to 22 causes and 20 risk factors from 2000 to 2019. The cut-off point of age 70 years was chosen because women in 164 and men in 165 of 186 countries and territories had a higher probability of dying before 70 years of age from an NCD than from communicable, maternal, perinatal

and nutritional conditions combined according to NCD Countdown 2030.[19 21]

## Statistical analysis

The population composition, change, dependency ratio, healthcare institution, personnel, and expenditures in terms of number, disease burden and risk factors in terms of number and rate were assessed by using a Joinpoint regression model. The disease burden and risk factor in individuals younger and older than 70 years in terms of number and rate were also assessed to identify the driving causes and risk factors. Average annual per cent change (AAPC) and associated 95% CIs were calculated for the study period. Temporal increasing and decreasing trends were defined according to the statistical significance of the AAPC compared with 0. AAPC values with 95% CIs overlapping with 0 were considered stable. Two segmented line regressions with joinpoint of 2000, 2009, and 2019 were calculated to show the temporal trend from 2000 to 2009, and from 2010 to 2019. All statistical analyses were performed by using the Joinpoint Regression Program (V.4.8.0.1, Statistical Methodology and Applications Branch, Surveillance Research Program, National Cancer Institute, Bethesda, Maryland, USA). P values less than 0.05 were considered statistically significant at a two-tailed level.

## Patient and public involvement

Patients and the public were not involved in any way in this research.

# RESULTS

## Population trends between 2000 and 2019

Initially, temporal trends of the population structure in China were examined based on CSY data in terms of gender and age composition, birth rate and dependency ratio. From 2000 through 2019, an increasing trend in total population (AAPC: 0.52, 95% CI: 0.51 to 0.54), male population (AAPC: 0.47, 95% CI: 0.44 to 0.51) and female population (AAPC: 0.59, 95% CI: 0.56 to 0.62) was observed. As expected, a strikingly contrasting trend in urban population (AAPC: 3.28, 95% CI: 3.18 to 3.38) and rural population (AAPC: −1.98, 95% CI: −2.02 to −1.95) was observed. We further identified that an increasing trend in urban population was attenuated, while a decreasing trend in rural population was aggravated, during 2010–2019 compared with 2000–2009 (figure 1A and table 1).

A clear decreasing trend in the younger population (ages 0–14 years) was observed in 2000–2019 (AAPC: −1.17, 95% CI: −1.78 to −0.55), especially in 2000–2009. An increasing trend in the working population (ages

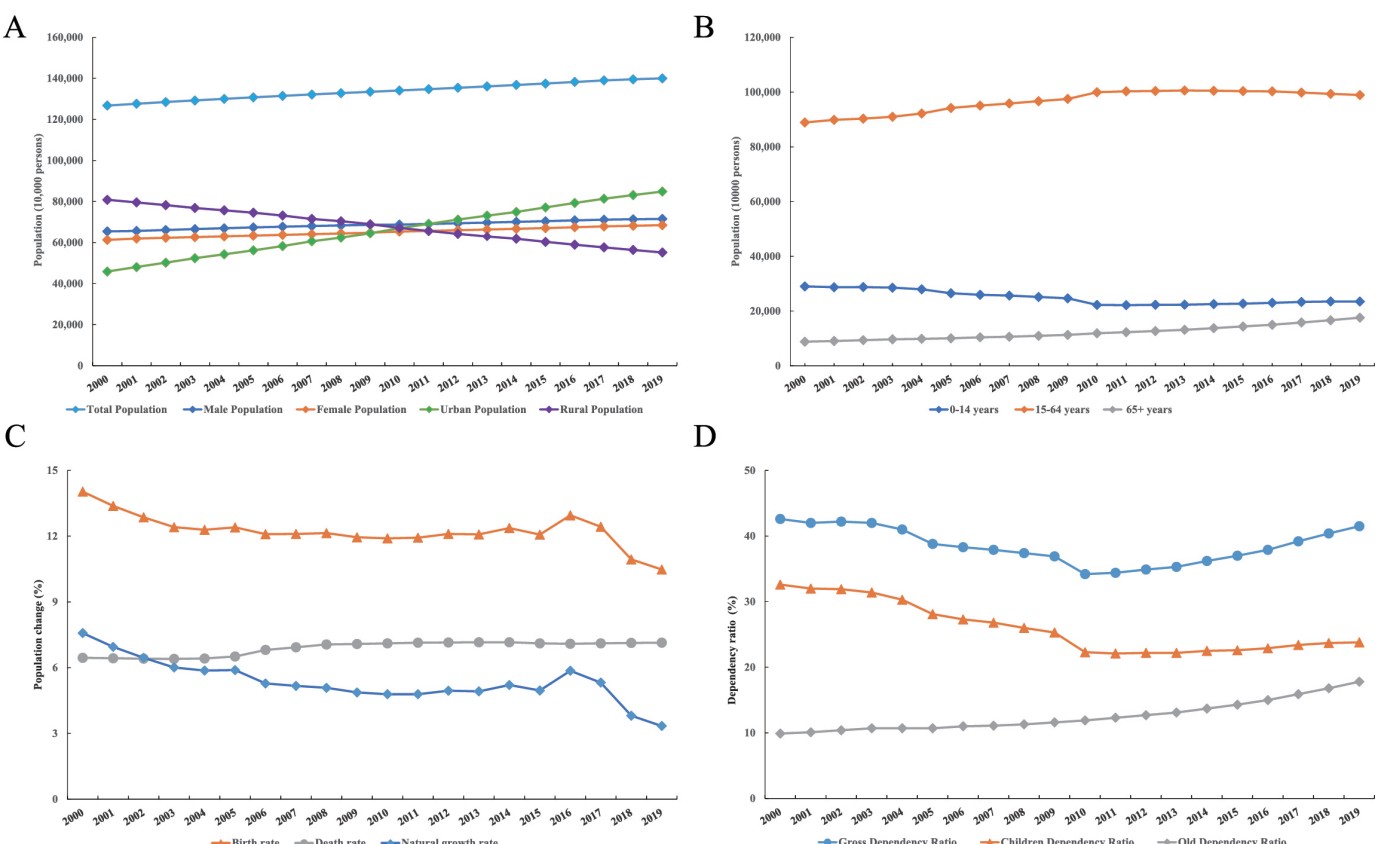

**Figure 1** Temporal trends of population in China between 2000 and 2019. Temporal trends of population composition are presented according to gender and area (A), age group (B), birth rate, death rate, and natural growth rate (C), and gross ratio, children ratio, and old dependency ratio (D).

**Table 1** The average annual per cent change (AAPC) in population composition, change and dependency ratio between 2000 and 2019

| Subcategory | Number | | | AAPC (95% CI) | | |
|---|---|---|---|---|---|---|
| | 2000 | 2009 | 2019 | 2000–2019 | 2000–2009 | 2010–2019 |
| Population composition by gender and area (10000 persons) | | | | | | |
| Total population | 126743 | 133450 | 140005 | 0.52 (0.51 to 0.54)* | 0.57 (0.56 to 0.58)* | 0.48 (0.45 to 0.51)* |
| Male population | 65437 | 68647 | 71527 | 0.47 (0.44 to 0.51)* | 0.53 (0.50 to 0.57)* | 0.43 (0.37 to 0.49)* |
| Female population | 61306 | 64803 | 68478 | 0.59 (0.56 to 0.62)* | 0.62 (0.56 to 0.67)* | 0.56 (0.55 to 0.57)* |
| Urban population | 45906 | 64512 | 84843 | 3.28 (3.18 to 3.38)* | 3.86 (3.75 to 3.98)* | 2.68 (2.51 to 2.84)* |
| Rural population | 80837 | 68938 | 55162 | −1.98 (−2.02 to −1.95)* | −1.79 (−1.86 to −1.73)* | −2.15 (−2.18 to −2.12)* |
| Population composition by age group (10000 persons) | | | | | | |
| Population aged 0–14 years | 29012 | 24659 | 23492 | −1.17 (−1.78 to −0.55)* | −2.24 (−2.76 to −1.71)* | 0.31 (−0.23 to 0.85) |
| Population aged 15–64 years | 88910 | 97484 | 98910 | 0.59 (0.52 to 0.66)* | 1.10 (1.03 to 1.18)* | 0.02 (−0.10 to 0.14) |
| Population aged 65+ years | 8821 | 11307 | 17603 | 3.67 (3.51 to 3.82)* | 2.81 (2.66 to 2.96)* | 4.51 (4.25 to 4.76)* |
| Population change (%) | | | | | | |
| Birth rate | 14.03 | 11.95 | 10.48 | −1.53 (−2.31 to −0.75)* | −1.63 (−2.32 to −0.93)* | −1.57 (−3.03 to −0.09)* |
| Death rate | 6.45 | 7.08 | 7.14 | 0.54 (0.37 to 0.71)* | 1.10 (0.75 to 1.46)* | 0.03 (−0.06 to 0.13) |
| Natural growth rate | 7.58 | 4.87 | 3.34 | −4.20 (−5.52 to −2.87)* | −4.64 (−5.69 to −3.57)* | −4.40 (−6.93 to −1.80)* |
| Dependency ratio (%) | | | | | | |
| Gross dependency ratio | 42.6 | 36.9 | 41.5 | −0.25 (−0.54 to 0.03) | −1.97 (−2.26 to −1.68)* | 1.70 (1.21 to 2.19)* |
| Children dependency ratio | 32.6 | 25.3 | 23.8 | −1.76 (−2.54 to −0.96)* | −3.33 (−4.00 to −2.66)* | 0.33 (−0.37 to 1.03) |
| Old dependency ratio | 9.9 | 11.6 | 17.8 | 3.12 (2.93 to 3.31)* | 1.76 (1.45 to 2.08)* | 4.49 (4.27 to 4.70)* |

*AAPC significantly differs from 0 at a two-tailed level of 0.05.

15–64 years) was observed in 2000–2009 (AAPC: 1.10, 95% CI: 1.03 to 1.18), and then it gradually disappeared over 2010–2019. In contrast, an increasing trend in the post-working population (65+ years) that was observed in 2000–2019 (AAPC: 3.67, 95% CI: 3.51 to 3.82) gradually intensified from 2000–2009 to 2010–2019 (figure 1B and table 1).

From 2000 to 2019, the birth rate (AAPC: −1.53, 95% CI: −2.31 to −0.75) and natural growth rate (AAPC: −4.20, 95% CI: −5.52 to −2.87) continued to decrease. Conversely, from 2000 to 2009, the death rate continued to increase (AAPC: 1.10, 95% CI: 0.75 to 1.46), yet from 2010 to 2019, it remained stable (AAPC: 0.03, 95% CI: −0.06 to 0.13) (figure 1C and table 1). The past two decades also exhibited a reverse trend in gross dependency ratio from 2010 to 2019, a muted decreasing trend in children dependency ratio from 2000–2009 to 2010–2019, and a strengthened increasing trend in old dependency ratio from 2000–2009 to 2010–2019 (figure 1D and table 1).

### Health resources and expenditure between 2000 and 2019
Temporal trends in health sources, including institutions, personnel and expenditures, were examined based on CSY data. A consistent increasing trend in number of hospitals (AAPC: 3.97, 95% CI: 3.69 to 4.24), including general hospitals, hospitals that specialise in traditional Chinese medicine and specialised hospitals, was observed. The number of medical personnel (AAPC: 3.26, 95% CI: 2.76 to 3.77), including medical technical personnel,

licensed assistant doctors, licensed doctors and registered nurses, also continued to increase in 2000–2019. Moreover, the rate of increasing number of medical personnel was higher in 2010–2019 than in 2000–2009. The number of beds in healthcare institutions (AAPC: 5.54, 95% CI: 5.20 to 5.88), including hospitals, township health service centres, specialised public health institutions, and women and children care agencies, also increased in 2000–2019. Like medical personnel, the rate increase for beds in healthcare institutions was greater in 2010–2019 than in 2000–2009 (table 2).

Total healthcare expenditure increased from ¥458.66 billion (US$55.40 billion) in 2000 to ¥6584.14 billion (US$954.68 billion) in 2019, with an AAPC of 15.28 (95% CI: 14.64 to 15.91). The per capita health expenditure increased from ¥361.88 in 2000 to ¥4702.79 in 2019, with an AAPC of 14.67 (95% CI: 14.04 to 15.30). Striking increasing trends in government health expenditure (AAPC: 18.95, 95% CI: 16.72 to 21.22), social health expenditure (AAPC: 19.03, 95% CI: 17.56 to 20.52) and out-of-pocket health expenditure (AAPC: 10.62, 95% CI: 9.24 to 12.02) were also observed in 2000–2019 (table 2).

### Population health, disease burden and risk factors between 2000 and 2019
According to data from GBD 2019, life expectancy at birth and at 60–64 years was estimated to increase (AAPC: 0.44, 95% CI: 0.43 to 0.46; AAPC: 0.76, 95% CI: 0.60 to 0.91, respectively). In addition, the percentage of healthy

**Table 2** The average annual per cent change (AAPC) in healthcare institution, personnel and expenditures between 2000 and 2019

| Subcategory | Number | | | AAPC (95% CI) | | |
|---|---|---|---|---|---|---|
| | 2000 | 2009 | 2019 | 2000–2019 | 2000–2009 | 2010–2019 |
| **Healthcare institution (unit)** | | | | | | |
| Number of hospitals | 16318 | 20291 | 34354 | 3.97 (3.69 to 4.24)* | 2.41 (2.03 to 2.80)* | 5.72 (5.25 to 6.19)* |
| Number of general hospitals | 11872 | 13364 | 19963 | 2.84 (2.56 to 3.13)* | 1.37 (0.97 to 1.77)* | 4.51 (4.03 to 4.99)* |
| Number of hospitals specialising in traditional Chinese medicine | 2453 | 2728 | 4221 | 2.92 (2.50 to 3.34)* | 1.35 (1.20 to 1.50)* | 4.69 (3.82 to 5.58)* |
| Number of specialised hospitals | 1543 | 3716 | 8531 | 9.65 (9.02 to 10.28)* | 10.98 (9.69 to 12.28)* | 8.46 (8.25 to 8.68)* |
| Number of healthcare institutions at grass roots level | 1000169 | 882153 | 954390 | −0.45 (−0.82 to −0.08)* | −1.88 (−2.60 to −1.15)* | 0.85 (0.67 to 1.04)* |
| Number of township health service centres | 49229 | 38475 | 36112 | −1.63 (−1.83 to −1.44)* | −2.79 (−3.10 to −2.47)* | −0.47 (−0.66 to −0.28)* |
| Number of village clinics | 709458 | 632770 | 616094 | −1.03 (−1.40 to −0.65)* | −1.88 (−2.53 to −1.23)* | −0.51 (−0.86 to −0.17)* |
| Number of outpatient departments | 240934 | 182448 | 266659 | 0.42 (−0.18 to 1.03) | −2.73 (−3.45 to −2.01)* | 4.05 (3.03 to 5.07)* |
| Number of specialised public health institutions | 11386 | 11665 | 15958 | 1.63 (−0.23 to 3.54) | 0.12 (−0.68 to 0.93) | 3.35 (−0.52 to 7.36) |
| Number of Centers for Disease Control and Prevention | 3741 | 3536 | 3403 | −0.40 (−0.50 to −0.29)* | −0.40 (−0.50 to −0.29)* | −0.40 (−0.50 to −0.29)* |
| Number of specialised disease prevention and treatment institutions | 1839 | 1291 | 1128 | −2.47 (−2.97 to −1.98)* | −3.61 (−4.46 to −2.76)* | −1.49 (−2.05 to −0.93)* |
| Number of women and children care agencies | 3163 | 3020 | 3071 | −0.11 (−0.29 to 0.08) | −0.45 (−0.80 to −0.10)* | 0.20 (0.09 to 0.32)* |
| **Medical personnel (10000 persons)** | | | | | | |
| Number of medical personnel | 691.04 | 778.14 | 1292.83 | 3.26 (2.76 to 3.77)* | 1.15 (0.16 to 2.14)* | 5.16 (4.87 to 5.45)* |
| Number of medical technical personnel | 449.08 | 553.51 | 1015.40 | 4.34 (3.99 to 4.68)* | 2.30 (1.61 to 3.01)* | 6.20 (6.07 to 6.34)* |
| Number of licensed (assistant) doctors | 207.58 | 232.92 | 386.69 | 3.12 (2.64 to 3.59)* | 0.92 (0.05 to 1.81)* | 5.36 (4.98 to 5.74)* |
| Number of licensed doctors | 160.33 | 190.54 | 321.05 | 3.53 (3.12 to 3.95)* | 1.50 (0.73 to 2.27)* | 5.62 (5.29 to 5.95)** |
| Number of registered nurses | 126.68 | 185.48 | 444.50 | 6.79 (6.20 to 7.38)* | 4.37 (3.26 to 5.49)* | 8.92 (8.49 to 9.34)* |
| Number of pharmacists | 41.44 | 34.19 | 48.34 | 0.76 (0.23 to 1.29)* | −2.28 (3.31 to −1.23)* | 3.57 (3.27 to 3.88)* |
| Number of village doctors and assistants | 131.94 | 105.10 | 84.23 | −2.49 (−4.28 to −0.66)* | −3.44 (−6.25 to −0.55)* | −2.16 (−4.10 to −0.17)* |
| Number of other technical personnel | 15.75 | 27.50 | 50.39 | 6.30 (6.07 to 6.53)* | 6.30 (6.07 to 6.53)* | 6.30 (6.07 to 6.53)* |
| Number of administrative personnel | 42.68 | 36.27 | 54.38 | 1.25 (0.42 to 2.09)* | −1.74 (−3.27 to −0.18)* | 4.01 (3.49 to 4.54)* |
| Number of logistics technical workers | 51.59 | 55.77 | 88.43 | 2.95 (2.13 to 3.78)* | 0.84 (−0.26 to 1.94) | 4.71 (3.57 to 5.86)* |
| **Beds in healthcare institutions** | | | | | | |
| Number of beds in healthcare institutions (10000 units) | 317.70 | 441.66 | 880.70 | 5.54 (5.20 to 5.88)* | 3.60 (3.01 to 4.20)* | 7.11 (6.82 to 7.41)* |
| Number of beds in hospitals (10000 beds) | 216.67 | 312.08 | 686.65 | 6.35 (5.96 to 6.74)* | 4.17 (3.55 to 4.81)* | 8.16 (7.74 to 8.59)* |

Continued

**Table 2** Continued

| Subcategory | Number | | | AAPC (95% CI) | | |
|---|---|---|---|---|---|---|
| | 2000 | 2009 | 2019 | 2000–2019 | 2000–2009 | 2010–2019 |
| Number of beds in community health service centres (10 000 units) | 76.65 | 109.98 | 163.11 | 3.96 (2.90 to 5.04)* | 3.73 (1.68 to 5.83)* | 3.28 (2.75 to 3.82)* |
| Number of beds in township health service centres (10 000 beds) | 73.48 | 93.34 | 136.99 | 3.29 (2.18 to 4.41)* | 2.38 (0.26 to 4.54)* | 3.48 (2.93 to 4.04)* |
| Number of beds in specialised public health institutions (10 000 units) | 11.86 | 15.40 | 28.50 | 4.80 (4.31 to 5.28)* | 2.97 (2.52 to 3.42)* | 6.24 (5.55 to 6.93)* |
| Number of beds in women and children care agencies (10 000 units) | 7.12 | 12.61 | 24.32 | 6.79 (6.41 to 7.18)* | 6.65 (6.15 to 7.15)* | 6.74 (6.25 to 7.24)* |
| Number of beds in specialised disease prevention and treatment institutions (10 000 beds) | 2.84 | 2.71 | 4.11 | 2.29 (0.21 to 4.41)* | 0.19 (-3.08 to 3.57) | 3.75 (1.61 to 5.95)* |
| **Health expenditure (¥100 million)** | | | | | | |
| Total health expenditure | 4586.63 | 17 541.92 | 65 841.39 | 15.28 (14.64 to 15.91)* | 15.99 (15.09 to 16.90)* | 14.04 (13.44 to 14.63)* |
| Government health expenditure | 709.52 | 4816.26 | 18 016.95 | 18.95 (16.72 to 21.22)* | 24.39 (20.91 to 27.97)* | 13.52 (11.30 to 15.79)* |
| Social health expenditure | 1171.94 | 6154.49 | 29 150.57 | 19.03 (17.56 to 20.52)* | 21.13 (18.16 to 24.18)* | 17.17 (16.18 to 18.16)* |
| Out-of-pocket health expenditure | 2705.17 | 6571.16 | 18 673.87 | 10.62 (9.24 to 12.02)* | 9.82 (9.21 to 10.44)* | 10.96 (8.48 to 13.48)* |
| Per capita health expenditure (¥) | 361.88 | 1314.26 | 4702.79 | 14.67 (14.04 to 15.30)* | 15.34 (14.45 to 16.24)* | 13.47 (12.89 to 14.06)* |

*AAPC significantly differs from 0 at a two-tailed level of 0.05.

life expectancy within total life expectancy at birth and at 60–64 years decreased from 88.90 to 88.32, and from 77.12 to 76.36, respectively (online supplemental figure 1 and tables 1–3). Overall, the number of all-cause DALYs was stable during 2000–2019 (AAPC: 0.08, 95% CI −0.06 to 0.22). However, it decreased between 2000 and 2009 (AAPC: −0.44, 95% CI: −0.58 to −0.31), and it increased between 2010 and 2019 (AAPC: 0.60, 95% CI: 0.39 to 0.82) (online supplemental table 4).

In 2000, the top three causes of DALYs were cardiovascular diseases, neoplasms and chronic respiratory diseases. In 2019, they were cardiovascular diseases, neoplasms and musculoskeletal disorders. From 2000 to 2019, the top three drivers of increasing number of DALYs were HIV/AIDS and sexually transmitted infections (AAPC: 3.47, 95% CI: 2.72 to 4.22), diabetes and kidney diseases (AAPC: 2.30, 95% CI: 2.09 to 2.51), and sense organ diseases (AAPC: 2.14, 95% CI: 1.98 to 2.30). The top two drivers of increasing age-standardised rate of DALYs were HIV/AIDS and sexually transmitted infections (AAPC: 2.02, 95% CI: 1.50 to 2.53) and musculoskeletal disorders (AAPC: 0.16, 95% CI: 0.10 to 0.23) (online supplemental table 5).

In 2000, the top three risk factors were tobacco, air pollution and high systolic blood pressure. In 2019, they were tobacco, high systolic blood pressure and dietary risks. Between 2000 and 2019, the top three leading risk factors with the largest increasing number of DALYs were high body mass index (AAPC: 4.02, 95% CI: 3.74 to 4.29), low physical activity (AAPC: 3.54, 95% CI: 2.70 to 4.38) and low bone mineral density (AAPC: 3.39, 95% CI: 3.12 to 3.67). The top three leading factors with the largest increase in age-standardised rate of DALYs were unsafe sex (AAPC: 1.47, 95% CI: 0.88 to 2.06), high body mass index (AAPC: 1.28, 95% CI: 0.95 to 1.60) and low bone mineral density (AAPC: 0.40, 95% CI: 0.27 to 0.54) (online supplemental table 6).

### Trends in disease burden between younger individuals and those older than 70 years

Finally, trends in causes and risk factors for the population with a cut-off point of 70 years were examined for 2000–2019. Among individuals younger than 70 years, musculoskeletal disorders replaced other NCDs as one of the top three causes from 2000 to 2019. In contrast, the top three causes remained the same among individuals older than 70 years, and included cardiovascular diseases,

neoplasms and chronic respiratory disease (table 3). Among the population younger than 70 years, the causes with the largest increment in number of DALYs were HIV/AIDS and sexually transmitted infections (AAPC: 3.33, 95% CI: 2.58 to 4.08), musculoskeletal disorders (AAPC: 1.88, 95% CI: 1.84 to 1.92) and sense organ diseases (AAPC: 1.71, 95% CI: 1.43 to 2.00). In the same population, the largest increments in rate of DALYs were HIV/AIDS and sexually transmitted infections (AAPC: 3.42, 95% CI: 2.76 to 4.09), musculoskeletal disorders (AAPC: 2.04, 95% CI: 2.00 to 2.08), and diabetes and kidney diseases (AAPC: 1.88, 95% CI: 1.58 to 2.18). In comparison, the causes among individuals older than 70 years that had the largest increment in both number and rate of DALYs were maternal and neonatal disorders (AAPC: 10.64, 95% CI: 10.47 to 10.82; AAPC: 6.84, 95% CI: 6.69 to 6.98, respectively), HIV/AIDS and sexually transmitted infections (AAPC: 8.17, 95% CI: 6.93 to 9.43; AAPC:4.39, 95% CI: 3.26 to 3.53, respectively) and unintentional injuries (AAPC: 4.66, 95% CI: 3.90 to 5.43; AAPC: 1.01, 95% CI: 0.46 to 1.56, respectively) (table 3).

Among the population younger than 70 years, dietary risks (AAPC: 1.02, 95% CI: 0.85 to 1.19) and high systolic blood pressure (AAPC: 1.55, 95% CI: 1.29 to 1.81) replaced air pollution and child and maternal malnutrition in the top three risk factors from 2000 to 2019. In contrast, the top three risk factors remained the same between 2000 and 2019 for the population older than 70 years, and they include: high systolic blood pressure, tobacco and air pollution (table 3). Regarding risk factors among the population younger than 70 years, the largest increments in number and rate of DALYs were high body mass index (AAPC: 3.70, 95% CI: 3.59 to 3.82; AAPC: 3.88, 95% CI: 3.69 to 4.07, respectively), unsafe sex (AAPC: 3.11, 95% CI: 2.80 to 3.42; AAPC: 3.28, 95% CI: 2.96 to 3.60, respectively) and low physical activity (AAPC: 2.75, 95% CI: 2.34 to 3.15; AAPC: 2.91, 95% CI: 2.46 to 3.36, respectively). In contrast, the risk factors with the largest increment in number of DALYs among those older than 70 years were high body mass index (AAPC: 5.22, 95% CI: 4.79 to 5.65), low bone mineral density (AAPC: 5.22, 95% CI: 4.98 to 5.46) and high low-density lipoprotein (LDL) cholesterol (AAPC: 5.02, 95% CI: 4.62 to 5.41). Furthermore, the same risk factors with different rankings were observed in terms of rate of DALYs for the same population: low bone mineral density (AAPC: 1.62, 95% CI: 1.38 to 1.86), high body mass index (AAPC: 1.50, 95% CI: 1.01 to 2.00) and high LDL cholesterol (AAPC: 1.44, 95% CI: 1.04 to 1.84), respectively (table 4).

## DISCUSSION

In the past two decades, although China invested hugely on health institution, personnel and expenditures to extend life expectancy and improve healthy life expectancy, population structure deteriorated, the total disease burden remained largely unchanged and age-driven disease burden continuously intensified. The trends

we observed in the population structure of China highlight the potential for an irreversible population decline to occur before the household wealth of G7 nations is achieved in China. Despite easing of birth limits in China, couples have been put off by high costs of living, expensive childcare, career choices and maternity leave. In addition, the post-working population has been increasing at an accelerating rate, suggesting a clear deterioration in population structure and ageing.[19] Thus, the ability of China to ease strains on its ageing population in the next 30 years is a pressing concern.[22]

On the one hand, continuous effort is needed to address the top three causes and risk factors which have been identified among the population in China between 2000 and 2019. Our findings show that neoplasms and cardiovascular diseases remain among the top three causes, and tobacco remains among the top three risk factors for individuals younger than 70 years. Comprehensive regional-specific strategies, including effective tobacco control policy, recommendations for healthier lifestyles and control of chronic infections, are needed as the cancer spectrum in China is changing from that of a developing country to that of a developed country.[23 24] Regulatory reforms also need to further reduce the lag between approval of new cancer drugs by the US Food and Drug Administration and their subsequent approval in China, and they need to facilitate the development and registration of drugs that may have clinical superiority over existing drugs.[25] Turning the inverted pyramid of the healthcare system is essential for the battle against cardiovascular diseases due to the increasing number of older patients, the high proportion of out-of-hospital deaths, and the gaps in lifestyle indicators between what is preferred and what is currently observed.[26 27] All provinces in mainland China have made great progress in increasing life expectancy at birth. Life course management of disease across the entire spectrum of healthcare services has led to a shift from a patient-centred and treatment-dominated model to a people-centred and health-centred model. It is hypothesised that the latter may help to integrate preventive and treatment measures for cardiovascular disease.[28] Tobacco sales have also increased despite a tight anti-smoking policy in China. Potential interference from the tobacco industry and the social currency of tobacco are suspected to be factors.[29–31] Therefore, great efforts should be taken not only to minimise these practices, but also to slow tobacco production.[32]

On the other hand, it is crucial to focus on incremental driving causes and risk factors in the early stages to mitigate future age-driven disease burden. Our data show that musculoskeletal disorders, dietary risks and high systolic blood pressure are among the top three causes and risk factors in people younger than 70 years. Split analyses were performed to examine variations according to age, and a cut-off point of 70 years was used because ischaemic heart disease and stroke were the top-ranked causes of DALYs in the population aged 70 years

**Table 3** The average annual per cent change (AAPC) of 22 causes in people younger and older than 70 years of age presented in disability-adjusted life years

| Causes | Age <70 years | | | Age ≥70 years | | |
|---|---|---|---|---|---|---|
| | 2000 | 2019 | AAPC (95% CI) | 2000 | 2019 | AAPC (95% CI) |
| **Number** | | | | | | |
| Neglected tropical diseases and malaria | 1 566 614 (1 022 581 to 2 343 199) | 977 312 (586 629 to 1 582 219) | −2.42 (−2.73 to −2.10)* | 137 451 (93 607 to 195 272) | 160 494 (103 985 to 236 535) | 0.83 (0.72 to 0.94)* |
| Nutritional deficiencies | 3 912 001 (2 872 239 to 5 269 234) | 2 247 485 (1 515 113 to 3 202 132) | −2.91 (−3.07 to −2.76)* | 271 587 (210 115 to 352 577) | 359 440 (285 475 to 450 281) | 1.51 (1.21 to 1.81)* |
| Neoplasms | 43 633 030 (40 123 940 to 47 258 486) | 48 503 754 (40 915 702 to 56 689 779) | 0.54 (0.36 to 0.73)* | 10 782 364 (10 021 201 to 11 549 415) | 19 016 076 (16 369 289 to 21 712 270) | 3.00 (2.66 to 3.34)* |
| Cardiovascular diseases | 40 746 813 (37 734 622 to 44 374 198) | 45 823 157 (39 092 553 to 53 248 142) | 0.67 (0.52 to 0.82)* | 26 837 679 (24 915 174 to 29 373 891) | 46 109 965 (40 669 589 to 51 698 469) | 2.77 (2.26 to 3.29)* |
| Chronic respiratory diseases | 13 237 789 (11 103 026 to 14 475 351) | 8 969 697 (7 748 010 to 10 534 163) | −2.04 (−2.22 to −1.86)* | 15 416 177 (12 665 862 to 16 567 447) | 13 550 852 (11 816 464 to 16 036 186) | −0.70 (−1.15 to −0.24)* |
| Digestive diseases | 9 775 769 (8 808 397 to 10 835 090) | 7 710 658 (6 445 828 to 8 964 870) | −1.21 (−1.30 to −1.13)* | 1 923 302 (1 762 799 to 2 106 969) | 2 307 223 (2 029 697 to 2 600 797) | 0.92 (0.72 to 1.13)* |
| Neurological disorders | 9 677 839 (4 755 522 to 16 786 166) | 11 207 269 (5 201 198 to 19 881 074) | 0.78 (0.70 to 0.86)* | 2 828 463 (1 694 636 to 5 326 904) | 6 174 136 (3 629 888 to 11 661 806) | 4.20 (4.04 to 4.35)* |
| Mental disorders | 17 241 666 (12 746 572 to 22 736 338) | 18 539 346 (13 738 299 to 24 152 654) | 0.37 (0.29 to 0.46)* | 868 363 (645 857 to 1 119 766) | 1 754 481 (1 291 689 to 2 254 704) | 3.77 (3.70 to 3.83)* |
| Musculoskeletal disorders | 17 725 280 (12 654 675 to 23 783 378) | 25 286 535 (18 045 649 to 33 942 087) | 1.88 (1.84 to 1.92)* | 2 364 272 (1 679 407 to 3 221 901) | 4 647 389 (3 292 799 to 6 332 328) | 3.60 (3.53 to 3.66)* |
| Other non-communicable diseases | 23 430 651 (19 446 807 to 28 355 357) | 16 912 189 (12 820 141 to 22 584 969) | −1.67 (−1.77 to −1.58)* | 1 142 340 (831 796 to 1 540 224) | 2 213 884 (1 599 002 to 2 992 458) | 3.43 (2.88 to 3.98)* |
| Skin and subcutaneous diseases | 7 427 502 (4 833 763 to 10 987 503) | 7 564 844 (4 920 805 to 11 324 882) | 0.09 (0.07 to 0.12)* | 381 523 (263 385 to 542 001) | 699 858 (475 558 to 1 030 747) | 3.21 (3.14 to 3.28)* |
| Sense organ diseases | 7 881 599 (5 268 499 to 11 424 474) | 10 796 982 (7 099 138 to 15 924 830) | 1.71 (1.43 to 2.00)* | 2 859 183 (2 030 582 to 3 892 635) | 5 184 684 (3 650 080 to 7 090 135) | 3.18 (3.02 to 3.34)* |
| Transport injuries | 16 117 112 (14 766 378 to 18 356 266) | 12 597 412 (10 747 152 to 14 456 584) | −1.31 (−1.64 to −0.98)* | 514 733 (454 228 to 601 574) | 1 148 043 (946 296 to 1 360 515) | 4.24 (3.92 to 4.57)* |
| Unintentional injuries | 19 929 305 (18 307 020 to 21 760 124) | 13 173 922 (10 853 663 to 15 202 200) | −2.15 (−2.65 to −1.65)* | 1 184 747 (1 024 676 to 1 371 585) | 2 829 540 (2 151 923 to 3 469 812) | 4.66 (3.90 to 5.43)* |
| Self-harm and interpersonal violence | 10 574 276 (9 275 669 to 11 544 266) | 5 232 715 (4 479 867 to 6 200 044) | −3.64 (−3.94 to −3.34)* | 597 000 (504 957 to 649 633) | 637 335 (544 995 to 738 168) | 0.37 (−0.73 to 1.50) |
| HIV/AIDS and sexually transmitted infections | 904 578 (711 182 to 1 201 186) | 1 677 944 (1 372 709 to 2 020 540) | 3.33 (2.58 to 4.08)* | 16 867 (13 900 to 22 250) | 74 476 (55 158 to 94 754) | 8.17 (6.93 to 9.43)* |
| Respiratory infections and tuberculosis | 19 691 768 (18 048 674 to 21 296 455) | 4 601 349 (3 952 381 to 5 376 466) | −7.32 (−7.79 to −6.85)* | 2 246 454 (2 045 162 to 2 412 569) | 2 046 523 (1 783 057 to 2 379 652) | −0.54 (−0.96 to −0.13)* |
| Enteric infections | 3 259 938 (2 859 853 to 3 751 832) | 1 179 711 (874 430 to 1 534 794) | −5.11 (−5.55 to −4.66)* | 147 328 (108 525 to 198 192) | 228 387 (164 251 to 306 599) | 2.47 (2.25 to 2.69)* |
| Other infectious diseases | 4 467 414 (3 610 475 to 6 110 768) | 1 143 025 (971 599 to 1 357 011) | −6.85 (−7.29 to −6.42)* | 114 576 (103 794 to 126 392) | 101 074 (86 237 to 116 157) | −0.59 (−0.98 to −0.21)* |
| Maternal and neonatal disorders | 18 100 148 (16 534 223 to 19 650 436) | 7 080 589 (6 220 729 to 8 094 637) | −4.86 (−5.45 to −4.27)* | 10 442 (7 450 to 14 363) | 70 983 (51 665 to 96 401) | 10.64 (10.47 to 10.82)* |

Continued

**Table 3** Continued

| Causes | Age <70 years | | | Age ≥70 years | | |
|---|---|---|---|---|---|---|
| | 2000 | 2019 | AAPC (95% CI) | 2000 | 2019 | AAPC (95% CI) |
| Substance use disorders | 6 144 467 (4 909 131 to 7 567 428) | 5 459 715 (4 162 774 to 6 975 972) | −0.55 (−0.97 to −0.13)* | 147 447 (122 722 to 177 177) | 295 943 (226 537 to 375 653) | 3.80 (3.23 to 4.37)* |
| Diabetes and kidney diseases | 8 018 292 (6 937 163 to 9 207 068) | 10 979 432 (9 048 531 to 13 121 983) | 1.70 (1.42 to 1.98)* | 2 418 531 (2 116 442 to 2 752 813) | 4 929 741 (4 274 671 to 5 652 658) | 3.78 (3.60 to 3.96)* |
| **Rate** | | | | | | |
| Neglected tropical diseases and malaria | 33 (21 to 49) | 21 (13 to 34) | −2.26 (−2.56 to −1.96)* | 250 (170 to 355) | 149 (96 to 219) | −2.70 (−2.80 to −2.59)* |
| Nutritional deficiencies | 82 (60 to 110) | 48 (33 to 69) | −2.77 (−2.96 to −2.58)* | 494 (383 to 642) | 333 (264 to 417) | −2.03 (−2.30 to −1.76)* |
| Neoplasms | 911 (838 to 987) | 1045 (881 to 1221) | 0.68 (0.59 to 0.77)* | 19 629 (18 243 to 21 025) | 17 613 (15 162 to 20 110) | −0.56 (−0.93 to −0.19)* |
| Cardiovascular diseases | 851 (788 to 927) | 987 (842 to 1147) | 0.84 (0.68 to 1.00)* | 48 857 (45 357 to 53 474) | 42 708 (37 669 to 47 885) | −0.69 (−0.82 to −0.57)* |
| Chronic respiratory diseases | 276 (232 to 302) | 193 (167 to 227) | −1.86 (−2.02 to −1.69)* | 28 064 (23 058 to 30 160) | 12 551 (10 945 to 14 853) | −4.13 (−4.60 to −3.66)* |
| Digestive diseases | 204 (184 to 226) | 166 (139 to 193) | −1.06 (−1.15 to −0.96)* | 3501 (3209 to 3836) | 2137 (1880 to 2409) | −2.55 (−2.66 to −2.45)* |
| Neurological disorders | 202 (99 to 351) | 241 (112 to 428) | 0.94 (0.82 to 1.06)* | 5149 (3085 to 9697) | 5719 (3362 to 10,801) | 0.59 (0.29 to 0.90)* |
| Mental disorders | 360 (266 to 475) | 399 (296 to 520) | 0.55 (0.51 to 0.59)* | 1581 (1176 to 2038) | 1625 (1196 to 2088) | 0.15 (0.12 to 0.18)* |
| Musculoskeletal disorders | 370 (264 to 497) | 545 (389 to 731) | 2.04 (2.00 to 2.08)* | 4304 (3057 to 5865) | 4305 (3050 to 5865) | −0.04 (−0.15 to 0.08) |
| Other non-communicable diseases | 489 (406 to 592) | 364 (276 to 486) | −1.54 (−1.66 to −1.42)* | 2080 (1514 to 2804) | 2051 (1481 to 2772) | −0.16 (−0.60 to 0.29) |
| Skin and subcutaneous diseases | 155 (101 to 229) | 163 (106 to 244) | 0.26 (0.23 to 0.29)* | 695 (479 to 987) | 648 (440 to 955) | −0.38 (−0.39 to −0.36)* |
| Sense organ diseases | 165 (110 to 239) | 233 (153 to 343) | 1.86 (1.70 to 2.02)* | 5205 (3697 to 7086) | 4802 (3381 to 6567) | −0.43 (−0.63 to −0.23)* |
| Transport injuries | 337 (308 to 383) | 271 (231 to 311) | −1.15 (−1.49 to −0.81)* | 937 (827 to 1095) | 1063 (876 to 1260) | 0.64 (0.33 to 0.96)* |
| Unintentional injuries | 416 (382 to 454) | 284 (234 to 327) | −1.93 (−2.44 to −1.42)* | 2157 (1865 to 2497) | 2621 (1993 to 3214) | 1.01 (0.46 to 1.56)* |
| Self-harm and interpersonal violence | 221 (194 to 241) | 113 (96 to 134) | −3.49 (−3.80 to −3.18)* | 1087 (919 to 1183) | 590 (505 to 684) | −3.10 (−4.05 to −2.14)* |
| HIV/AIDS and sexually transmitted infections | 19 (15 to 25) | 36 (30 to 44) | 3.42 (2.76 to 4.09)* | 31 (25 to 41) | 69 (51 to 88) | 4.39 (3.26 to 3.53)* |
| Respiratory infections and tuberculosis | 411 (377 to 445) | 99 (85 to 116) | −7.17 (−7.61 to −6.71)* | 4090 (3723 to 4392) | 1896 (1652 to 2204) | −4.08 (−4.44 to −3.71)* |
| Enteric infections | 68 (60 to 78) | 25 (19 to 33) | −4.91 (−5.76 to −4.05)* | 268 (198 to 361) | 212 (152 to 284) | −1.20 (−1.45 to −0.96)* |
| Other infectious diseases | 93 (75 to 128) | 25 (21 to 29) | −6.70 (−7.14 to −6.27)* | 209 (189 to 230) | 94 (80 to 108) | −4.06 (−4.48 to −3.64)* |
| Maternal and neonatal disorders | 378 (345 to 410) | 153 (134 to 174) | −4.71 (−5.29 to −4.14)* | 19 (14 to 26) | 66 (48 to 89) | 6.84 (6.69 to 6.98)* |
| Substance use disorders | 128 (103 to 158) | 118 (90 to 150) | −0.38 (−0.85 to 0.09)* | 268 (223 to 323) | 274 (210 to 348) | 0.12 (−0.31 to 0.55) |
| Diabetes and kidney diseases | 167 (145 to 192) | 236 (195 to 283) | 1.88 (1.58 to 2.18)* | 4403 (3853 to 5011) | 4566 (3959 to 5236) | 0.21 (−0.05 to 0.48) |

*AAPC significantly differs from 0 at a two-tailed level of 0.05.

**Table 4** The average annual per cent change (AAPC) of 20 risk factors in people younger and older than 70 years of age

| Risk factors | Age <70 years | | | Age ≥70 years | | |
|---|---|---|---|---|---|---|
| | 2000 | 2019 | AAPC (95% CI) | 2000 | 2019 | AAPC (95% CI) |
| **Number** | | | | | | |
| Unsafe water, sanitation and handwashing | 3 996 889 (3 183 257 to 4 887 734) | 923 837 (645 583 to 1 254 723) | −7.42 (−8.15 to −6.69)* | 237 825 (168 202 to 320 654) | 228 667 (159 219 to 307 023) | −0.14 (−0.42 to 0.13) |
| Air pollution | 32 546 257 (28 559 720 to 36 241 262) | 24 301 291 (20 582 581 to 28 569 480) | −1.52 (−1.70 to −1.34)* | 16 789 702 (14 415 333 to 19 203 189) | 18 208 412 (15 565 447 to 21 260 363) | 0.27 (0.00 to 0.54)* |
| Other environmental risks | 3 596 355 (2 485 008 to 4 876 076) | 3 222 783 (1 977 748 to 4 651 395) | −0.53 (−0.77 to −0.29)* | 1 790 347 (1 199 126 to 2 480 222) | 3 083 288 (2 034 326 to 4 259 059) | 2.86 (2.48 to 3.23)* |
| Child and maternal malnutrition | 27 473 911 (24 705 249 to 30 472 454) | 6 937 292 (6 012 640 to 7 958 846) | −6.98 (−7.79 to −6.16)* | 256 375 (197 956 to 332 924) | 316 491 (248 797 to 398 362) | 1.15 (0.66 to 1.63)* |
| Tobacco | 37 227 472 (33 809 017 to 40 999 710) | 38 743 936 (32 232 707 to 46 296 710) | 0.22 (0.08 to 0.37)* | 20 037 114 (18 407 452 to 21 694 478) | 25 372 064 (22 095 893 to 28 843 357) | 1.24 (0.79 to 1.69)* |
| Alcohol use | 12 593 606 (10 822 456 to 14 698 783) | 13 876 946 (11 135 736 to 16 941 170) | 0.47 (0.39 to 0.56)* | 2 079 979 (1 635 218 to 2 555 460) | 3 388 114 (2 514 663 to 4 384 540) | 2.51 (2.10 to 2.92)* |
| Drug use | 5 880 735 (5 044 491 to 6 863 455) | 4 596 666 (3 725 262 to 5 556 055) | −1.24 (−1.46 to −1.02)* | 420 543 (338 823 to 505 234) | 525 128 (420 389 to 638 546) | 1.10 (0.84 to 1.35)* |
| High fasting plasma glucose | 10 850 938 (8 940 115 to 13 134 425) | 16 082 215 (12 638 442 to 19 989 086) | 2.09 (1.69 to 2.50)* | 7 073 207 (5 239 551 to 9 435 756) | 12 146 224 (9 068 765 to 16 392 373) | 2.86 (2.43 to 3.29)* |
| High systolic blood pressure | 21 853 891 (18 103 552 to 25 435 298) | 28 919 385 (23 316 926 to 34 533 463) | 1.55 (1.29 to 1.81)* | 13 487 603 (10 597 533 to 16 167 287) | 25 522 231 (20 992 213 to 30 524 269) | 3.37 (2.90 to 3.84)* |
| High body mass index | 9 311 731 (3 191 606 to 17 435 563) | 18 520 264 (9 033 132 to 29 514 190) | 3.70 (3.59 to 3.82)* | 2 392 758 (706 129 to 5 015 786) | 6 309 777 (2 532 862 to 11 224 097) | 5.22 (4.79 to 5.65)* |
| Low bone mineral density | 1 234 815 (1 026 536 to 1 443 017) | 1 913 892 (1 512 289 to 2 304 790) | 2.37 (2.10 to 2.64)* | 534 562 (448 070 to 635 290) | 1 406 383 (1 058 504 to 1 740 813) | 5.22 (4.98 to 5.46)* |
| Dietary risks | 24 107 231 (19 699 737 to 29 721 184) | 29 061 110 (22 436 447 to 36 581 781) | 1.02 (0.85 to 1.19)* | 10 146 090 (7 449 653 to 13 767 645) | 17 752 016 (12 683 501 to 23 791 732) | 2.90 (2.59 to 3.21)* |
| Low physical activity | 479 307 (211 814 to 1 031 049) | 793 859 (332 155 to 1 622 549) | 2.75 (2.34 to 3.15)* | 795 916 (391 872 to 1 491 328) | 1 713 262 (787 433 to 3 275 090) | 3.97 (3.34 to 4.60)* |
| Occupational risks | 14 894 928 (12 651 846 to 17 338 599) | 11 828 821 (9 596 301 to 14 324 948) | −1.20 (−1.31 to −1.10)* | 3 676 403 (2 901 359 to 4 413 363) | 3 658 028 (2 941 788 to 4 424 853) | −0.10 (−0.45 0.25) |
| Intimate partner violence | 766 126 (417 803 to 1 191 498) | 739 392 (335 422 to 1 209 468) | −0.12 (−0.36 to 0.12)* | 23 715 (10 039 to 44 392) | 50 534 (16 543 to 99 799) | 4.12 (3.96 to 4.29)* |
| Unsafe sex | 1 352 963 (1 174 728 to 1 769 992) | 2 445 381 (1 761 748 to 2 945 569) | 3.11 (2.80 to 3.42) | 129 732 (113 048 to 182 721) | 279 416 (197 910 to 338 222) | 4.29 (3.79 to 4.79)* |
| Non-optimal temperature | 3 641 568 (2 714 325 to 4 594 010) | 3 468 758 (2 670 613 to 4 405 174) | −0.24 (−1.11 to 0.64) | 4 785 750 (4 234 955 to 5 404 845) | 5 856 605 (4 952 561 to 6 867 429) | 0.76 (0.49 to 1.02)* |
| Kidney dysfunction | 6 244 259 (5 571 754 to 7 041 339) | 7 536 042 (6 221 117 to 9 011 471) | 1.08 (0.82 to 1.35)* | 2 639 944 (2 227 097 to 3 100 279) | 5 817 866 (4 717 831 to 7 022 861) | 4.30 (3.63 to 4.97)* |
| High LDL cholesterol | 8 217 261 (6 919 451 to 9 686 198) | 12 184 167 (9 541 238 to 14 971 019) | 2.16 (1.91 to 2.42)* | 3 000 527 (1 868 209 to 4 589 459) | 7 629 795 (4 751 976 to 11 248 818) | 5.02 (4.62 to 5.41)* |

Continued

**Table 4** Continued

| Risk factors | Age <70 years | | | Age ≥70 years | | |
|---|---|---|---|---|---|---|
| | 2000 | 2019 | AAPC (95% CI) | 2000 | 2019 | AAPC (95% CI) |
| Childhood sexual abuse and bullying | 755 840 (307 189 to 1 434 919) | 629 932 (269 309 to 1 156 818) | −0.95 (−1.06 to −0.84)* | 6905 (2954 to 12 651) | 16 924 (7261 to 32 044) | 4.86 (4.72 to 5.00)* |
| Rate | | | | | | |
| Unsafe water, sanitation and handwashing | 83 (66 to 102) | 20 (14 to 27) | −7.24 (−7.97 to −6.51)* | 433 (306 to 584) | 212 (147 to 284) | −3.66 (−3.92 to −3.40)* |
| Air pollution | 680 (596 to 757) | 523 (443 to 615) | −1.36 (−1.53 to −1.19)* | 30 565 (26 242 to 34 958) | 16 865 (14 417 to 19 692) | −3.04 (−3.35 to −2.74)* |
| Other environmental risks | 75 (52 to 102) | 69 (43 to 100) | −0.37 (−0.59 to −0.14)* | 3259 (2183 to 4515) | 2856 (1884 to 3945) | −0.64 (−0.95 to −0.32)* |
| Child and maternal malnutrition | 574 (516 to 636) | 149 (130 to 171) | −6.83 (−7.48 to −6.18)* | 467 (360 to 606) | 293 (230 to 369) | −2.43 (−2.80 to −2.06)* |
| Tobacco | 777 (706 to 856) | 835 (694 to 997) | 0.38 (0.23 to 0.53)* | 36 476 (33 510 to 39 494) | 23 500 (20 466 to 26 716) | −2.29 (−2.71 to −1.87)* |
| Alcohol use | 263 (226 to 307) | 299 (240 to 365) | 0.61 (0.33 to 0.90)* | 3786 (2977 to 4652) | 3138 (2329 to 4061) | −1.09 (−1.57 to −0.61)* |
| Drug use | 123 (105 to 143) | 99 (80 to 120) | −1.02 (−1.22 to −0.82)* | 766 (617 to 920) | 486 (389 to 591) | −2.34 (−2.54 to −2.15)* |
| High fasting plasma glucose | 227 (187 to 274) | 346 (272 to 431) | 2.25 (1.82 to 2.69)* | 12 876 (9538 to 17 177) | 11 250 (8400 to 15 183) | −0.79 (−1.48 to −0.10)* |
| High systolic blood pressure | 456 (378 to 531) | 623 (502 to 744) | 1.72 (1.47 to 1.98)* | 24 553 (19 292 to 29 432) | 23 639 (19 444 to 28 272) | −0.15 (−0.42 to 0.13) |
| High body mass index | 194 (67 to 364) | 399 (195 to 636) | 3.88 (3.69 to 4.07)* | 4356 (1285 to 9131) | 5844 (2346 to 10 396) | 1.50 (1.01 to 2.00)* |
| Low bone mineral density | 26 (21 to 30) | 41 (33 to 50) | 2.54 (2.20 to 2.89)* | 973 (816 to 1157) | 1303 (980 to 1612) | 1.62 (1.38 to 1.86)* |
| Dietary risks | 503 (411 to 621) | 626 (483 to 788) | 1.19 (1.02 to 1.36)* | 18 470 (13 562 to 25 063) | 16 442 (11 748 to 22 037) | −0.58 (−0.74 to −0.43)* |
| Low physical activity | 10 (4 to 22) | 17 (7 to 35) | 2.91 (2.46 to 3.36)* | 1449 (713 to 2715) | 1587 (729 to 3033) | 0.40 (−0.21 to 1.02)* |
| Occupational risks | 311 (264 to 362) | 255 (207 to 309) | −1.05 (−1.10 to −0.99)* | 6693 (5282 to 8034) | 3388 (2725 to 4098) | −3.51 (−3.97 to −3.05)* |
| Intimate partner violence | 16 (9 to 25) | 16 (7 to 26) | 0.02 (−0.23 to 0.27)* | 43 (18 to 81) | 47 (15 to 92) | 0.49 (0.31 to 0.66)* |
| Unsafe sex | 28 (25 to 37) | 53 (38 to 63) | 3.28 (2.96 to 3.60)* | 236 (206 to 333) | 259 (183 to 313) | 0.69 (0.16 to 1.22)* |
| Non-optimal temperature | 76 (57 to 96) | 75 (58 to 95) | −0.05 (−0.89 to 0.81)* | 8712 (7710 to 9839) | 5425 (4587 to 6361) | −2.59 (−3.60 to −1.56)* |
| Kidney dysfunction | 130 (116 to 147) | 162 (134 to 194) | 1.25 (1.00 to 1.49)* | 4806 (4054 to 5644) | 5389 (4370 to 6505) | 0.64 (−0.02 to 1.31) |
| High LDL cholesterol | 172 (144 to 202) | 262 (206 to 322) | 2.33 (2.07 to 2.59)* | 5462 (3401 to 8355) | 7067 (4401 to 10 419) | 1.44 (1.04 to 1.84)* |
| Childhood sexual abuse and bullying | 16 (6 to 30) | 14 (6 to 25) | −0.80 (−0.89 to −0.72)* | 13 (5 to 23) | 16 (7 to 30) | 1.21 (1.06 to 1.37)* |

*AAPC significantly differs from 0 at a two-tailed level of 0.05.
LDL, low-density lipoprotein.

and older.[19] A previous study quantified the temporal trends in disease burden of musculoskeletal disorders which could be attributed to changes in risk factors.[27] The results obtained support population-wide initiatives which target high body mass index to help mitigate the burden of musculoskeletal disorders.[33] Changes in diets to correct for urbanisation-driven dietary shifts to higher consumption of refined sugars, refined fats, oils and meats may also help prevent diet-related and chronic NCDs.[34] With resources available to control high systolic blood pressure and reduce the burden of hypertension, efforts are needed to improve access to better antihypertensive medications and to implement artificial intelligence-based digital health.[35 36]

We identified key domains that require intervention in order to reduce an increasing disease burden among individuals older than 70 years. Most importantly, the top three causes and risk factors for this age group remained unchanged. Thus, our findings indicate that greater effort is needed to tackle the rapidly increasing age-standardised rate of DALYs due to traditional major risk factors, including high body mass index, low bone mineral density and high LDL cholesterol. Moreover, the increasing burden of maternal and neonatal disorders, HIV/AIDS and sexually transmitted infections, and unintentional injuries needs to be highlighted as well.

Healthcare for older individuals is undergoing unprecedented challenges due to underprepared geriatric hospitals and nursing homes, an unbalanced healthcare insurance system between urban and rural areas, and an absence of education on national ageing and post-working health.[22] Consequently, there is an urgent need to identify traits in older people who experience short healthspan. By identifying healthy behaviours and casual genetic and metabolic factors of the latter, it may be possible to decrease the burden of late-life diseases and extend healthspan.[37] The development of long-term health systems to meet the needs of older individuals in an age-friendly environment is fundamental to fostering healthy ageing and maximising the functional abilities of older individuals.[38]

Healthcare has experienced an upsurge regarding medical institutions, personnel, beds and expenditures. Our findings clearly demonstrate that China has increased its health spending much faster than its economic growth. In 2018, China provided $644.7 million for the development of health assistance.[39] This trend in increasing healthcare expenditures will be further aggravated considering the high costs related to chronic illnesses, population growth and ageing in the USA.[9 40] While China has reformed its primary healthcare system to insure its population,[7 41] improving the quality of primary healthcare and increasing reimbursement for chronic NCDs are key steps to alleviating national disease burden.[42 43] In addition, the cost-effectiveness of health expenditures on disease burden and targeting the poor should be primary considerations in public health policy given the limited financial resources for health.[44]

## Limitation of this study

It must also be noted that there were several limitations associated with the present study. First, our findings might be biased by changes in the age structure of the population and different populations over a specified period because temporal trends in population and healthcare utilisation were not quantified in terms of age-standardised rate. Second, population and healthcare data from CSY 2020 were combined with modelled disease burden data from GBD 2019 based on an assumption that these two sources of data are consistent. Third, the most recent trends could not be captured due to time lags in data release by the NBS of China and GBD studies. Fourth, all the limitations of the GBD methodology described elsewhere[3 19 20] could have potentially affected the present study. Fifth, owing to the nature of the data and methods from CSY, we are currently unable to parse population estimates into more narrow age groups and provide deflator-adjusted health expenditure data. National-level data were collected and analysed, and therefore, within-country variations were not considered.

## CONCLUSION

Comprehensive age/cause/risk factor-specific strategies are key to reconcile the tension among the triad of population ageing, disease burden and healthcare expenditure. The disease burden from cardiometabolic diseases, neoplasms and musculoskeletal disorders was identified as key domains that require intervention to reduce an increasing disease burden among individuals currently older than 70 years, as well as those approaching this age group.

**Contributors** Prof. JG is the guarantor of the study and had full access to all the data in the study and takes responsibility for the integrity of the data and the accuracy of the data analysis. ZL, DW, CG and JG conceived and designed the study, performed the analysis, and wrote the paper. All authors read and commented on the manuscript and approved the final version of the manuscript. The corresponding author attests that all listed authors meet authorship criteria and that no others meeting the criteria have been omitted.

**Funding** Project funded by China Postdoctoral Science Foundation (2020TQ0380) and Guangdong Clinical Research Center of Immune Disease (2020B1111170008). The Global Burden of Disease Study is funded by the Bill and Melinda Gates Foundation.

**Disclaimer** The funder was not involved in the preparation of this manuscript.

**Competing interests** None declared.

**Patient and public involvement** Patients and/or the public were not involved in the design, or conduct, or reporting, or dissemination plans of this research.

**Patient consent for publication** Not required.

**Ethics approval** Data released from the Global Health Data Exchange query and the China Statistical Yearbook did not require informed patient consent. This study used an anonymised publicly available dataset with no identifiable information of the survey participants. Thus, ethics approval was not required for this study.

**Provenance and peer review** Not commissioned; externally peer reviewed.

**Data availability statement** Data are available in a public, open access repository. Data used for the analyses are publicly available from the Institute of Health Metrics and Evaluation (https://ghdx.healthdata.org/gbd-2019;https://ghdx.healthdata. org/gbd-2019/data-input-sources) and the National Bureau of Statistics of China

**ORCID iDs**
Zhenguo Liang http://orcid.org/0000-0001-6356-7731
Dongze Wu http://orcid.org/0000-0002-5571-8728

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
