## [Reviewer comments · BMJ Open]

ARTICLE DETAILS

TITLE (PROVISIONAL)	Temporal trend of population structure, burden of diseases, healthcare resources and expenditure in China, 2000-2019
AUTHORS	Liang, Zhenguo; WU, Dongze; Guo, Cui; Gu, Jieruo

VERSION 1 – REVIEW

REVIEWER	Ling Sun Nanjing University Medical School
REVIEW RETURNED	12-Mar-2022

GENERAL COMMENTS	Comments on bmjopen-2022-062091 This manuscript described the trends in population and health care using data from China Statistical Yearbook and assessed the disease burden using GBD data, in China during 2000-2019. It does not provide much more new information for each aspect alone, but does help readers to understand when comparing three metrics at a level. I have some comments as followed. 1 The authors should strengthen in describing previous related studies and compare with them. What have been reported about each subject and what does this study added? For example, Maigeng Zhou et al has published a series of articles to detail the disease burden and risk factors in China and even at the provincial level. 2 Please detailed the method and conditions for calculating AAPC by the Joinpoint Regression Analysis. 3 Have the authors considered the changes of age structure and different population over a specified period, when estimating the trend in numbers and unstandardized rate. It means the AAPC should be calculated based standardized values.
--

REVIEWER	Kun Zou Sichuan University West China Second University Hospital
REVIEW RETURNED	17-Mar-2022

GENERAL COMMENTS	The authors aimed to examine the trends of population structure, main population health indicators, burden of diseases and contributed risk factors, to inform future health policy and health resource allocation in China. It is applaudable that the authors take a perspective of population based medicine to address the health of the public in China. It is also delighted to see the emphasis on prevention and control of
---

	main population risk factors, and rational allocation of health resources according to the disease/health priorities of the public, which is especially valuable as they came from our colleagues of general medicine. However, there are several concerns.  1. The title did not reflect the main content of this study exactly. It might be “Secular change of population structure, burden of diseases, healthcare resources and expenditure in China, 2000-2019”. 2. The introduction did not manage to give a clear background and justification of this research. It would be good to reorganise the introduction to fully justify the need of this study. 3. Methods, China Statistic Yearbook and Global burden of disease can be put under a subtitle “Data sources” 4. Methods, another section named “Outcomes of measurement” would be appreciated to clarify all the outcomes being used in this study. It would be even better to organise these outcomes into several classifications. 5. Methods, acronym is normally not used in titles or subtitles, such as that of the two databases. 6. Methods, Statistical analysis, the authors may give more details to clarify the methods used for each outcome of measurement. Besides, statistics used for subgroup or stratified analysis, such as that by age groups, should also be explicitly described. 7. There is room for improvement as regard to academic writing, including words phrases, tense, and logic of paragraphs and sentences. 8. Results, Healthcare trends between 2000 and 2009, might be “Health resources and expenditure” , and may be put after the “Trends in causes and risk factors between 2000 and 2019” 9. Results, “Trends in causes and risk factors between 2000 and 2019” might be “Population health, disease burden and risk factors” 10. It interesting that “The causes among individuals older than 70 years that had the largest increment in both number and rate of DALYs were maternal and neonatal disorders”, is there any possible explanation? 11. The discussion is a strength of the paper. However, it would be better if the authors could summarise all the main findings of the study in the first paragraph of Discussion. Thereafter, point-by-point discussion can be made for each main finding in the following paragraphs, within context of relevant literature. 12. It is good to have a paragraph of “limitations of this study”. However, some of the points make could not be well understood as limitations, which may need more clarification. 13. Conclusions could benefit from more specific implications for public health interventions(policies) and future research.
--	---

VERSION 1 – AUTHOR RESPONSE

Reviewer: 1

Question #1:

The authors should strengthen in describing previous related studies and compare with them. What have been reported about each subject and what does this study added. For example, Maigeng Zhou et al has published a series of articles to detail the disease burden and risk factors in China and even at the provincial level.

Response#1:

Response #1a: Thank you for your valuable suggestion. We have added following sentence into the introduction part: "Although two GBD studies have assessed the disease burden and risk factor at provincial level in China ((PMID: 31248666, 26510778)), the study further to evaluate healthcare professional and expenditure and to identify key domain to mitigate the aging driven disease burden." (introduction section, line84-86, page4)

Response #1b: We have also added following sentence into second paragraph of the introduction part. "The study also tries to dissect the implication of a central tension among 3 mutually conflicting forces: intractable disease burden, population ageing, and upsurging healthcare investment." (introduction section, line86-88, page4)

Question #2:

Please detailed the method and conditions for calculating AAPC by the Joinpoint Regression Analysis.

Response#2a:

The method and conditions for calculating AAPC by the Joinpoint Regression Analysis have been detailed described as follows.

The sentence "Temporal trends in population, health care, and disease burden were assessed by using a Joinpoint regression model." were revised into the sentence "The population composition, change, dependency ratio, healthcare institution, personnel, and expenditures in terms of number, disease burden and risk factors in terms of number and rate, were assessed by using a Joinpoint regression model." (method section, line121-123, page6)

Response#2b:

The following sentence was added into method section to explicitly describe subgroup analysis of age. "The disease burden and risk factor in individuals younger and older than 70 years in term of number and rate were also assessed to identify the driving cause and risk factors." (method section , line123-125, page6)

Response#2c:

The following sentence was added into method section. "Two segmented line regression with joinpoint of 2000, 2009, and 2019 were calculated to show the temporal trend from 2000 to 2009, from 2010 to 2019." (method section, line128-130, page6)

Question #3:

Have the authors considered the changes of age structure and different population over a specified period, when estimating the trend in numbers and unstandardized rate. It means the AAPC should be calculated based standardized values.

Response#3:

Thank you very much for pointing out this. The China Statistical Yearbook only reported population and healthcare data in term of number. And we have calculated AAPC of disease burden and risk factors in term of age standardized rate.

We have revised the first limitation in last paragraph of discussion part.

"First, temporal trends in health care utilization, risk factors, and disease burden were not quantified due to changes in the age structure of the population and temporal trends in disease burden attributable to changes in risk factors."

was revised into

"First, our findings might be biased by changes in the age structure of the population and different population over a specified period because temporal trends in population and health care utilization were not quantified in terms of age standardized rate." (discussion section, line338-341, page23)

Reviewer: 2

Question #1:

The title did not reflect the main content of this study exactly. It might be “Secular change of population structure, burden of diseases, healthcare resources and expenditure in China, 2000-2019”.

Response#1:

Thank you for your great idea. We have revised the title into “Temporal trend of population structure, burden of diseases, healthcare resources and expenditure in China, 2000-2019”.

Question #2:

The introduction did not manage to give a clear background and justification of this research. It would be good to reorganize the introduction to fully justify the need of this study.

Response #2a:

Thank you for your valuable suggestion. We have added following sentence into the introduction part: “Although two GBD studies have assessed the disease burden and risk factor at provincial level in China, the study further to evaluate healthcare professional and expenditure and to identify key domain to mitigate the aging driven disease burden.” (introduction section, line84-86, page4)

Response #2b:

We have also added following sentence into second paragraph of the introduction part. “The study also tries to dissect the implication of a central tension among 3 mutually conflicting forces: intractable disease burden, population ageing, and upsurging healthcare investment.” (introduction section, line86-88, page4)

Question #3:

Methods, China Statistic Yearbook and Global burden of disease can be put under a subtitle “Data sources”

Response#3:

The subtitle "Data sources" has been created to include both China Statistical Yearbook and the Global Burden of Disease. (methods section, line93, page5)

Question #4:

Methods, another section named “Outcomes of measurement” would be appreciated to clarify all the outcomes being used in this study. It would be even better to organize these outcomes into several classifications.

Response#4:

Thank you for your advice. The outcome of measurement section was created into method part as follows.

“Outcomes of measurement

The primary outcomes are temporal trend in population, healthcare, and disease burden from 2000 to 2019, including population composition, population change, dependency ratio, healthcare institution, personnel, expenditure, disease burden and risk factor. The secondary outcomes are i) the main drivers of disease burden and risk factor in individuals younger and older than 70, ii) temporal trend in population, healthcare, and disease burden from 2000 to 2009, and from 2010 to 2019.” (Method section, line113-119, page 6).

Question #5:

Methods, acronym is normally not used in titles or subtitles, such as that of the two databases.

Response#5:

Thank you for your advice. The subtitle "Data sources" has been created to include both China Statistical Yearbook and the Global Burden of Disease.

(Method section, line93, page5)

Question #6:

Methods, Statistical analysis, the authors may give more details to clarify the methods used for each outcome of measurement. Besides, statistics used for subgroup or stratified analysis, such as that by age groups, should also be explicitly described.

Response#6a:

The method and conditions for calculating AAPC by the Joinpoint Regression Analysis have been detailed described as follows.

The sentence “Temporal trends in population, health care, and disease burden were assessed by using a Joinpoint regression model.” were revised into the sentence “The population composition, change, dependency ratio, healthcare institution, personnel, and expenditures in terms of number, disease burden and risk factors in terms of number and rate, were assessed by using a Joinpoint regression model.” (method section, line121-123, page6)

Response#6b:

The following sentence was added into method section to explicitly describe subgroup analysis of age. “The disease burden and risk factor in individuals younger and older than 70 years in term of number and rate were also assessed to identify the driving cause and risk factors.” (method section , line123-125, page6)

Response#6c:

The following sentence was added into method section. “Two segmented line regression with joinpoint of 2000, 2009, and 2019 were calculated to show the temporal trend from 2000 to 2009, from 2010 to 2019.” (method section, line128-130, page6)

Response#6d:

The outcome of measurement section was created into method part as follows.

“Outcomes of measurements.

The primary outcomes are temporal trend in population, healthcare, and disease burden from 2000 to 2019, including population composition, population change, dependency ratio, healthcare institution, personnel, expenditure, disease burden and risk factor. The secondary outcomes are i) the main drivers of disease burden and risk factor in younger and older individuals, ii) temporal trend in population, healthcare, and disease burden from 2000 to 2009, and from 2010 to 2019.” (method section, line113-119, page6)

Question #7:

There is room for improvement as regard to academic writing, including words phrases, tense, and logic of paragraphs and sentences.

Response#7:

Thank you. A native speaker was invited to proofread the revised manuscript.

Question #8:

Results, Healthcare trends between 2000 and 2009, might be “Health resources and expenditure”, and may be put after the “Trends in causes and risk factors between 2000 and 2019”

Response#8:

Thank you for your valuable suggestion.

We have changed “Healthcare trends between 2000 and 2019” into “Health resources and expenditure between 2000 and 2019”. (results section, line161, page10)

We did not change place of the section because both population and healthcare data come from China Statistical Yearbook, while cause and risk factor data come from Global Burden of Disease study.

Question #9:

Results, “Trends in causes and risk factors between 2000 and 2019” might be “Population health, disease burden and risk factors”

Response#9:

Thank you for the advice. The “Trends in causes and risk factors between 2000 and 2019” has been revised into “Population health, disease burden and risk factors between 2000 and 2019”. (results section, line183, page13)

Question #10:

It interesting that “The causes among individuals older than 70 years that had the largest increment in both number and rate of DALYs were maternal and neonatal disorders”, is there any possible explanation?

Response#10:

According to Table 3, we learned that the maternal and neonatal disorders had the largest increment in both number and rate of DALYs among individuals older than 70 years. The result might be explained by the fact that in 2000, the number and rate of DALYs among individuals older than 70 years were quite small compare those younger than 70 years and were smallest in 22 causes among individuals older than 70 years.

Question#11:

The discussion is a strength of the paper. However, it would be better if the authors could summaries all the main findings of the study in the first paragraph of Discussion. Thereafter, point-by-point discussion can be made for each main finding in the following paragraphs, within context of relevant literature.

Response#11:

The following sentence was added into the start of first paragraph in discussion part.

“In the past two decades, although China invested huge health institution, personnel, and expenditures to extend life expectancy and improve healthy life expectancy, population structure deteriorated, the total disease burden remained largely unchanged, ageing driven disease burden continuously intensified.” (discussion section, line258-261, page20)

Question#12:

It is good to have a paragraph of “limitations of this study”. However, some of the points make could not be well understood as limitations, which may need more clarification.

Response#12:

Thank you for the note. The last paragraph summarized the limitation of this study. A subtitle of “Limitation of this study”were added into start of last of paragraph in discussion part. (discussion section, line337, page23)

Question#13:

Conclusions could benefit from more specific implications for public health interventions(policies) and future research.

Response#13:

Thank you. The conclusion has been revised as follows

“Comprehensive age-cause-risk factor-specific strategies that target the younger population in China are needed to mitigate the causes and risk factors which have paralleled dynamic population changes that have occurred between 2000 and 2019 in China. The disease burden from metabolic risk factors, neoplasms and musculoskeletal disorders were identified as key domains that require intervention to reduce an increasing disease burden among individuals currently older than 70 years, as well as those approaching this age group. There remain huge unmet clinical needs across different age and future study should focus on improve suboptimal quality of life for patients with new drugs and strategies.” (conclusion section, line353-360, page24)

VERSION 2 – REVIEW

REVIEWER	Kun Zou Sichuan University West China Second University Hospital
REVIEW RETURNED	05-Jul-2022

GENERAL COMMENTS	The manuscript has been improved. However, there are still several concerns. Major:  1. The necessity of this study has not fully justified by the introduction yet, more description and a summary on findings from exiting literature on population structure, burden of disease, healthcare resources and expenditure in China is needed; the limitations of current research also warrant explicit clarification accordingly, to support this study. 2. There have been research on trend of population structure, burden of disease, healthcare resources and expenditure in China, respectively. Why the combination of these elements became a strength of this study, as stated in the “Strengths and limitations of this study”? Could the authors explain more in the instruction or the discussion section? 3. The sentences and paragraphs may be restructured to improve the logic of the section of Introduction 4. Were aggregated or individualised data of NBS and GBD used in the analysis? The level of data used in the analysis needs clarification in the methods section. 5. Outcomes of measurement, do you mean “number of healthcare institutions”, “number of health personnel”? 6. The indicator of “disease burden” needs clarification. 7. What does “drivers of disease burden” mean? Causes of disease burden? 8. Outcomes of measurements and statistical analysis, why drivers of disease burden and risk factors were assessed in population younger and older than 70 years? Why an analysis of the entire population was not performed firstly? The reasons for these analyses warrant further clarification. 9. In the analysis of population composition, age group of 0-14, 15-64 and 65+ were used. However, in the analysis of disease burden, age group younger and older than 70 years were used. Why are they different? This may need further clarification and justification. 10. Minor:  1. Abstract, “setting” might be “data source”? 2. The conclusion in the abstract may be more specific and consistent with that in the main text. 3. There are still typos and grammar errors, another proof reading by a native English speaker may be needed. For example: “invested huge health institution, personnel...” might be “invested hugely on...”
--

VERSION 2 – AUTHOR RESPONSE

Reviewer: 2

Question #1:

The necessity of this study has not fully justified by the introduction yet, more description and a summary on findings from exiting literature on population structure, burden of disease, healthcare

resources and expenditure in China is needed; the limitations of current research also warrant explicit clarification accordingly, to support this study.

Response#1:

Thank you for the great advice. We have added the following sentence into the introduction part.

Ageing society in China continue to exacerbate the burden borne by current family and public healthcare systems, which make it is necessary to make long-term strategic plans to respond to the pressures of an ageing society at the governmental, individual, and technologic level [PMID:32971255]. There are a very low spatial and temporal matching degrees between the population-ageing rate and the number of medical resources per thousand residents in China, and the geographical pattern of the temporal matching between them exhibited a feature of north-south differentiation [PMID:32493251]. China does not have a typical disease characteristic of an aging society comparing the characteristics of the aging population in China with those in the world [PMID:33732678]. Namely, China faces the dual threat of non-communicable diseases and communicable diseases, and the former account for most of the age-related disease burden. Although there are great number of studies centering on trend of population, burden of disease, healthcare in China, but they failed to dissect the implication of a central tension among 3 mutually conflicting forces: intractable disease burden, population ageing, and upsurging healthcare expenditure." (Page 5. Line 90-91 and Page 6. Line 92-102)

Question #2:

There have been research on trend of population structure, burden of disease, healthcare resources and expenditure in China, respectively. Why the combination of these elements became a strength of this study, as stated in the "Strengths and limitations of this study"? Could the authors explain more in the instruction or the discussion section?

Response#2:

Thank you. We have added the following sentence into the introduction part.

"Although there are great number of studies centering on trend of population, burden of disease, healthcare in China, but they failed to dissect the implication of a central tension among 3 mutually conflicting forces: intractable disease burden, population ageing, and upsurging healthcare expenditure. The study tries to find a solution to reconcile the three contradictory forces, achieving longer healthy lifespan in the ageing society at high-cost performance of health expenditure." (Page 6. Line 99-104)

Question #3:

The sentences and paragraphs may be restructured to improve the logic of the section of Introduction.

Response#3a:

Thanks for your advice. A new paragraph was created and added into the introduction part.

Ageing society in China continue to exacerbate the burden borne by current family and public healthcare systems, which make it is necessary to make long-term strategic plans to respond to the pressures of an ageing society at the governmental, individual, and technologic level [PMID:32971255]. There are a very low spatial and temporal matching degrees between the

population-ageing rate and the number of medical resources per thousand residents in China, and the geographical pattern of the temporal matching between them exhibited a feature of north-south differentiation [PMID:32493251]. China does not have a typical disease characteristic of an aging society comparing the characteristics of the aging population in China with those in the world [PMID:33732678]. Namely, China faces the dual threat of non-communicable diseases and communicable diseases, and the former account for most of the age-related disease burden. Although there are great number of studies centering on trend of population, burden of disease, healthcare in China, but they failed to dissect the implication of a central tension among 3 mutually conflicting forces: intractable disease burden, population ageing, and upsurging healthcare expenditure. The study tries to find a solution to reconcile the three contradictory forces, achieving longer healthy lifespan in the ageing society at high-cost performance of health expenditure.” (Page 5. Line 90-91 and Page 6. Line 92-104)

Response#3b:

The last sentence in last paragraph of introduction part was also revised as follows.

“Possible driving forces of disease burden among ageing populations were also examined to provide a basis for targeted interventions, such as early disease prevention and health management among older populations.” (Page 6. Line 104-107)

Question #4:

Were aggregated or individualised data of NBS and GBD used in the analysis? The level of data used in the analysis needs clarification in the methods section.

Response#4:

Thank you. The following sentence was added into method section.

“Aggregated data of population and healthcare from CSY and aggregated data of life expectancy, disease burden attributable to 22 causes or 20 risk factors from GBD between 2000 and 2019 were used to quantify temporal trend of population structure, burden of diseases, healthcare resources and expenditure in China.” (Page 7. Line 129-132)

Question #5:

Outcomes of measurement, do you mean “number of healthcare institutions”, “number of health personnel”?

Response#5:

Yes. We have revised the following sentence to clearly clarify it.

“The primary outcomes are temporal trend in population (population composition, population change, dependency ratio), healthcare (number of healthcare institution, number of health personnel, healthcare expenditure), life expectancy (healthy and total), disease burden attributable to 22 causes or 20 risk factors (disability adjusted life years, DALYs) from 2000 to 2019. “(Page 7. Line 134-137)

Question #6:

The indicator of “disease burden” needs clarification.

Response#6a:

We have revised the following sentence to clearly clarify it.

“The primary outcomes are temporal trend in population (population composition, population change, dependency ratio), healthcare (number of healthcare institution, number of health personnel, healthcare expenditure), life expectancy (healthy and total), disease burden attributable to 22 causes or 20 risk factors (disability adjusted life years, DALYs) from 2000 to 2019.” (Page 6. Line 134-137)

Question #7:

What does “drivers of disease burden” mean? Causes of disease burden?

Response#7a:

Among 22 causes or 20 risk factors, driver was the cause or risk factor that have the greatest impact on the total disease burden attributable to 22 causes or 20 risk factors from 2000 to 2019.

The 22 causes include neglected tropical diseases and malaria, nutritional deficiencies, neoplasms, cardiovascular diseases, chronic respiratory diseases, digestive diseases, neurological disorders, mental disorders, musculoskeletal disorders, other non-communicable diseases, skin and subcutaneous diseases, sense organ diseases, transport injuries, unintentional injuries, self-harm and interpersonal violence, HIV/AIDS and sexually transmitted infections, respiratory infections and tuberculosis, enteric infections, other infectious diseases, maternal and neonatal disorders, substance use disorders, diabetes and kidney diseases.

The 20 risk factors include unsafe water, sanitation, and handwashing, air pollution, other environmental risks, child and maternal malnutrition, tobacco, alcohol use, drug use, high fasting plasma glucose, high systolic blood pressure, high body-mass index, low bone mineral density, dietary risks, low physical activity, occupational risks, intimate partner violence, unsafe sex, non-optimal temperature, kidney dysfunction, high LDL cholesterol, childhood sexual abuse and bullying

Response#7b:

We have revised the secondary outcomes to clearly indicate what is driver of disease burden.

“The secondary outcomes are i) the main drivers of disease burden from 22 causes or 20 risk factors in individuals younger and older than 70, ii) temporal trend in population, healthcare, and disease burden attributable to 22 causes or 20 risk factors from 2000 to 2009, and from 2010 to 2019. Driver refers the cause or risk factor that have the greatest impact on the total disease burden attributable to 22 causes or 20 risk factors from 2000 to 2019.” (Page 7, Line 137 and Page 8, Line 138-142)

Question #8:

Outcomes of measurements and statistical analysis, why drivers of disease burden and risk factors were assessed in population younger and older than 70 years? Why was an analysis of the entire population not performed firstly? The reasons for these analyses warrant further clarification.

Response#8:

Thank you. We have firstly analyzed the trend of disease burden attributable to 22 causes or 20 risk factors from 2000 to 2019 for the entire population (sTable 5, sTable 6). Please refer it in “Population health, disease burden and risk factors between 2000 and 2019” section.

Then we analyzed the trend of disease burden attributable to 22 causes or 20 risk factors from 2000 to 2019 in population younger and older than 70 years (Table 3, Table 4). Please refer it in “Trends in disease burden between younger individuals and those older than 70 years” section.

Question #9:

In the analysis of population composition, age group of 0-14, 15-64 and 65+ were used. However, in the analysis of disease burden, age group younger and older than 70 years were used. Why are they different? This may need further clarification and justification.

Response#9a:

The following sentence was added into Discussion part.

“Two set of age group was used to specifically assess the trend of population composition and disease burden. Population composition (0-14, 15-64, 65+) and dependency ratio (gross, children, old) were used to reflect the trend of working population and population ageing as age 0-14, 15-64 and 65+ group were younger population, working population, post-working population, respectively. The analysis of disease burden in population younger and older than 70 years aimed to identify the cause or risk factor (driver) which has the greatest impact on the trend of disease burden attributable to 22 causes and 20 risk factors from 2000 to 2019. The outpoint of age 70 was chosen because women in 164 and men in 165 of 186 countries and territories had a higher probability of dying before 70 years of age from a non-communicable disease (NCD) than from communicable, maternal, perinatal, and nutritional conditions combined according to NCD Countdown 2030. [PMID: 30264707, 33069326]” (Page 24. Line 358-369)

Please also refer to the definition of gross dependency ratio, children dependency ratio, and old dependency ratio, which was included in glossary and definitions of supplementary materials. Gross dependency ratio refers to the ratio of children aged 0-14 and elderly population aged 65 and over to the working-age population aged 15-64. It describes in general the number of non-working-age population that every 100 people at working ages will take care of. Children dependency ratio refers to the ratio of children to the working-age population. It describes the number of children that every 100 people of working age take care of. Old dependency ratio refers to the ratio of the elderly to the working-age population. It describes the number of elderlies that every 100 people of working age take care of. Old dependency ratio is one of the indicators that reflect the social implication of population aging from an economic perspective.

Question #10:

Abstract, “setting” might be “data source”?

Response#10

Thank you for the advice. We have revised “setting” into “data source”. (Page 2. Line 27)

Question #11:

The conclusion in the abstract may be more specific and consistent with that in the main text.

Response#11a:

Thank you for the advice.

We have revised the objective into “To explore the evolutionary trend of population structure, disease burden, healthcare resources and expenditure in China, and to identify key domains that are most in need of intervention.” (Page 2, Line 22-24)

Response#11b:

We have revised the conclusion into “Comprehensive age-cause-risk factor-specific strategies are key to reconcile the tension among triangle of population ageing, disease burden and healthcare expenditure. The disease burden from cardiometabolic diseases, neoplasms and musculoskeletal disorders were identified as key domains that require intervention to reduce an increasing disease burden among individuals currently older than 70 years, as well as those approaching this age group.” (Page 3. Line 49-53 and Page 25. Line 386-391)

Question #12:

There are still typos and grammar errors, another proof reading by a native English speaker may be needed. For example: “invested huge health institution, personnel...” might be “invested hugely on...”

Response#12

Thank you for your advice and point out this grammar error. A native English speaker has been invited to proofread the revised manuscript for typos and grammar errors.

The sentence has been revised as follows. “In the past two decades, although China invested hugely on health institution, personnel, and expenditures to extend life expectancy and improve healthy life expectancy, population structure deteriorated, the total disease burden remained largely unchanged, ageing driven disease burden continuously intensified.” (Page 21. Line 279-282)